# Involving Parents to Help Improve Children’s Energy Balance-Related Behaviours Through a School-Based Intervention

**DOI:** 10.3390/ijerph17134838

**Published:** 2020-07-05

**Authors:** Anke H. Verhees, Sacha R.B. Verjans-Janssen, Dave H.H. Van Kann, Stef P.J. Kremers, Steven B. Vos, Sanne M.P.L. Gerards

**Affiliations:** 1Department of Health Promotion, NUTRIM School of Nutrition and Translational Research in Metabolism, Maastricht University, 6229 HA Maastricht, The Netherlandss.kremers@maastrichtuniversity.nl (S.P.J.K.); sanne.gerards@maastrichtuniversity.nl (S.M.P.L.G.); 2School of Sport Studies, Fontys University of Applied Sciences, 5644 HZ Eindhoven, The Netherlands; steven.vos@fontys.nl; 3Department of Industrial Design, Eindhoven University of Technology, 5600 MB Eindhoven, The Netherlands

**Keywords:** children, intervention, nutrition, parental involvement, physical activity, school

## Abstract

The Challenge Me intervention aimed to indirectly involve parents in a school-based intervention, by challenging primary school children to perform physical activity (PA) and nutrition-related activities with their parents. The aim of this study is to gain insight in whether this was a feasible strategy to engage children and parents, especially those of vulnerable populations. An exploratory cross-sectional study design was applied. Four primary schools implemented the intervention. Data consisted of challenges completed (intervention posters) and child and family characteristics (questionnaires and anthropometric measurements). Associations between challenges performed and child and family characteristics were assessed using linear regression analysis. Of the 226 study participants, 100% performed at least one challenge, and 93% performed at least one challenge involving parents. Children who performed more PA challenges were often younger, a sports club member, lived in higher socioeconomic status neighbourhoods, of Western ethnicity and from larger families. Regarding nutrition challenges involving parents, younger children performed more challenges. There was no difference in intervention engagement regarding gender, weight status, PA preference, healthy nutrition preference, or the Family PA and Family Nutrition Climate. Challenge Me has potential in involving parents in a school-based intervention. However, certain characteristics were associated with higher involvement.

## 1. Introduction

Recent studies show a rise in overweight and obesity among children and adolescents to 18% worldwide in 2016 [1], with one in five school-aged children in Europe being overweight or obese [2,3]. Also in The Netherlands, childhood obesity rates are high. In 2019, 12% of children aged 4 to 12 were overweight or obese [4]. Overweight or obesity in childhood is especially problematic, since this often continues into adolescence and adulthood and is related to an increased risk of negative health consequences, such as type II diabetes, hypertension, respiratory disease and various types of cancer [2,3,5]. Overweight and obesity are preventable and treatable [6], especially at a young age [5,7], by improving healthy nutrition behaviours and increasing physical activity (PA) (also called energy balance-related behaviours (EBRBs)) [8,9,10]. To target children’s EBRBs [8,11], multiple settings [12], such as the school [13] and home environment should be involved in interventions [8,13].

Since children spend a large proportion of their weekdays at school, and schools reach many children, schools have been a popular intervention setting for decades [13,14,15]. However, the influence of the home setting on children’s EBRBs is profound, especially for young children, since parents determine the food availability at home, and influence children’s nutrition and PA behaviour by practices like modelling and rule setting [16,17,18,19]. Therefore it is important to also focus on the home setting when implementing school-based energy balance-related interventions [15,17].

Most effective changes in the home setting are accomplished through interventions with direct parental involvement (e.g., parents attending educational sessions, or counselling sessions) [13,16,19,20,21]. However, directly involving parents is often resource- and labour-intensive, making these types of interventions less feasible [20]. Also, the recruitment [22] and prolonged engagement of parents in these types of interventions have proven to be challenging [3,22,23]. Only about one-third of invited families participate in any intervention activity [3,24], 40 to 60% of whom drop out [24]. In addition, the participants of parental involvement interventions tend to be mostly high socioeconomic status (SES) parents [25], and it is particularly challenging to engage parents with a low SES [3,25], a lower educational level, single parents and those of ethnic minority groups [24]. This is discouraging, since these parents/families are most in need of interventions. For example, research from a large Dutch cohort study has shown that there are socioeconomic and ethnic inequalities in child health [26]. At a young age, non-Western children were more likely to be overweight compared to Dutch children, with mother’s educational level being one of the contributing factors in explaining this higher prevalence in overweight in these children [27]. Beyond socioeconomic health disparities, child characteristics and the supportiveness of the home environment to perform healthy behaviours are also important predictors of children’s PA and nutrition behaviour. For instance, children’s nutrition behaviour is determined by their preference for healthy and/or unhealthy foods, and their PA and sedentary behaviour is associated with their activity preferences [28]. Also, children raised in an environment in which healthy nutrition [29] and sufficient PA [29,30] are less valued are more at risk for developing unhealthy EBRBs. Therefore, vulnerability of the population is not restricted to well-known socioeconomic differences, but also includes specific child preferences and the health climate regarding EBRBs in the home environment.

To involve more parents, and particularly those of vulnerable populations, the strategy of indirect parental involvement is an alternative. Even though indirect parental involvement is assumed to be less effective in changing health behaviours compared to direct parental involvement, it can lead to greater adoption and implementation rates [21]. In indirect parental involvement, parents are engaged in a way that the intervention implementers do not communicate or engage directly (i.e., face to face, or personally) with them. Instead, parents are informed via school media, or children function as the messenger. Examples of indirect parental involvement are the provision of information (newsletters, tip sheets), invitations to participate in or attend activities (events, educational sessions) and prompts or assignments directed at the child and/or parent with the aim of involving parents [19]. Previous research has shown that some of these strategies are more promising than others, for example prompting children to engage in intervention activities together with their parent(s) seems more promising than providing newsletters or invitations for optional intervention activities [17,19]. The difference between these strategies could be described as passive (not requiring a specific action or response on the part of the parents) indirect involvement versus active (performing activities) indirect involvement. 

To our knowledge, there is little to no research on the participation rates of the strategy of the latter type of indirect parental involvement in school-based interventions, while this information is crucial to be able to draw conclusions on potential strategies to involve parents in (school-based) interventions [21]. Given the fact that children of vulnerable populations are less active, more sedentary and have unhealthier diets [31,32,33], it is possible that these children and their parents are less interested in energy balance-related interventions. Therefore, gaining insight in participation rates of children and parents in interventions using an indirect parental involvement strategy is needed. Moreover, empirical evidence is lacking on who is engaged in this strategy and therefore, it is warranted to know whether vulnerable children and parents engage in these interventions using this strategy.

In the current study, we evaluated the potential of Challenge Me to engage children and parents, especially those of vulnerable populations. Challenge Me was a parental involvement intervention in which children are challenged to perform PA and nutrition-related activities by themselves and with their parents. The main aim of the current study is to examine whether conducting challenges is a feasible intervention strategy to engage children and parents in school-based energy balance-related interventions. With engagement, we refer to participation in the intervention, i.e., performing the intervention activities, and not enrolment. Additionally, a second aim is to gain insight in whether children in need for improvements, i.e., vulnerable populations, are engaged in the intervention using this indirect-involvement strategy. For this aim, we focussed on child demographics, child characteristics (preferences) and the climate towards health behaviours in the home environment.

## 2. Materials and Methods 

### 2.1. Design

An exploratory cross-sectional study design was conducted. The Medical Ethics Committee of the Maastricht University Medical Centre and Maastricht University provided ethical approval for this study (METC163027, national number: NL58554.068.16).

### 2.2. Challenge Me Intervention

#### 2.2.1. Intervention Development

The intervention, called Challenge Me, challenged children to perform PA and nutrition-related challenges together with their parent(s) or guardian(s). Challenge Me has been developed as part of a larger evaluation [34], and is specifically focused on improving parental involvement. In the larger evaluation study, eight primary schools, located in low SES neighbourhoods (based on scoring of The Netherlands Institute for Social Research [35]) in a city in the south of The Netherlands, participated. During the development and implementation of Challenge Me in the local context, researchers collaborated with a local youth work organisation. This local organisation developed the challenges, with input from local health, education and sports professionals. Challenge Me consisted of several easy-to-perform activities, introduced weekly at school by people locally well-known by means of instruction videos (i.e., the schools’ own PE teacher, local free runners, a famous vlogger and a television chef from the region). A selection of the challenges was designed to involve parents. The local nature of the intervention enhanced a fit with the schools’ local context, which we anticipated would improve programme use, recognition and commitment by the children and consequently the involvement of parent(s).

The result of this design process was an intervention consisting of a total of eighteen challenges (Table 1). Challenges were categorised in ‘child-only’ (nine challenges performed by a child on his/her own, together with other children or in class) and ‘parental involvement’ (nine challenges that required parental assistance). Challenges could further be categorised as PA-focussed (e.g., ‘Run as far as possible with your entire class, in 2 min’/ ‘Go exercise for 30 min together with your parent(s)/guardian(s)’), or nutrition-focussed (e.g., ‘Find a healthy dinner recipe and bring this with you to school’/‘Make a healthy breakfast for someone in your household’). To ensure active involvement of the children and parents in the intervention, children had to provide proof of completing a challenge, which was specified per challenge (e.g., by taking a picture of their exercise). The class that performed the most challenges (corrected for the number of children in each class) was awarded a prize. The prize was a PA workshop, provided by the local youth work organisation.

The local youth work organisation distributed the materials. The materials included a Challenge Me poster on which the various challenges were listed (Appendix A, Figure A1), an instruction form for the teacher, stickers (used to mark a challenge as completed) and four instruction videos. The local youth organisation also appointed an employee specifically tasked with the implementation and monitoring of Challenge Me. This local youth worker also provided all necessary instructions in participating classes, to limit additional workload of teachers.

#### 2.2.2. Implementation

Recruitment for the intervention took place at school level. Intervention schools (*N* = 8) already participated in a larger evaluation study [34]. These schools were invited to implement the Challenge Me intervention. Of these, four school decided to participate. Four schools declined to participate because of time constraints. Teachers of participating schools were orally informed by the local youth worker about Challenge Me and its aims, and received instructions on how to perform the intervention. The same youth worker also informed the children about the intervention and monitored the progress of the intervention.

The participating schools implemented the intervention in grade six to eight (children aged 9 to 12), except for one. This school only implemented the intervention in grade eight. Each class received a Challenge Me poster. Challenges were expected to be performed during a four-week period (January–February 2019). Every week, the teacher showed one of the provided introduction videos provided by the local youth worker to kick-start the week. Depending on the challenge, challenges were performed at home, or in the neighbourhood. Children and parents participated in the intervention voluntarily. Based on the proof provided by the child, the teacher decided whether a challenge had been completed successfully. At the end of the four weeks, the local youth worker collected all the posters and decided which class had won.

### 2.3. Study Participants

Children from grades six to eight (aged 9 to 12) were eligible for inclusion in the study [34]. Parents received an informed consent form via the child [34]. In short, the informed consent form stated the purpose of the study, and how collected data would be handled. It also emphasised that participation in the study was voluntary, and that participants could always withdraw without stating any reason for doing so. For children to participate in the study, both parents had to provide written consent. This was based on the regulations of the Central Committee on Research involving Human Subjects [36].

### 2.4. Data Collection

For this exploratory study, data from the larger evaluation study were used [34]. Data were collected in March–April 2019, after the completion of the Challenge Me intervention in all four schools. During that period, both children and their parents received a questionnaire. Children received and filled in the questionnaire in the classroom, during regular school hours, with one researcher and at least one research assistant present to provide instructions. Parent questionnaires were handed out to the children in an envelope. Parents filled in the questionnaire at home, and returned it to school via the child. Children’s height and weight were measured within the same week as the child questionnaire.

#### 2.4.1. Performance of Challenges

The posters provided data on which challenges were completed by whom. Data were pseudonymised and digitalised. Challenges were grouped in various categories (Table 1)—PA: PA child-only and PA parental involvement; nutrition: nutrition child-only and nutrition parental involvement. Outcome measures were the percentage of challenges performed.

#### 2.4.2. Child Characteristics

Child characteristics were assessed via a questionnaire. Children reported their date of birth, gender (boy/girl), country of birth, country of birth of their mother and father, the number of brothers and sisters and whether they were a member of a sports club (yes/no). The child’s age was calculated based on the date of birth. In addition, Western or non-Western ethnicity of the child was determined, based on the country of birth of both parents. A child was considered as having a non-Western ethnicity when at least one parent was born in a non-Western country, based on the definition of Huntington [37]. The number of siblings was determined, based on the number of brothers and/or sisters.

Nutrition and PA preferences were also measured via a questionnaire, using an instrument developed by Rodenburg et al. [28]. This instrument ranks food, drink and leisure-time activity preferences by means of comparison. Items were visualized by means of an infographic of a child holding a food or drink item, or portraying an activity, accompanied by the description of the food, drink or activity. The items were compared in pairs, and children were asked to indicate the food, drink or activity of their preference. The food items were fruit, vegetables, sweet snacks (e.g., candy and cookies) and savoury snacks (e.g., crisps, nuts, cheese). The drinks items included sugar-sweetened beverages, light drinks (i.e., drinks that were artificially sweetened), tea without sugar, fruit juice and water (water was added to the items at a later stage). Leisure-time activity consisted of eight items, namely cycling, using the computer, watching television, playing sports, dancing, arts and crafts, reading and playing outside. In total, the instrument consisted of 44 comparisons. Items were re-categorised into active activities (cycling, playing sports, dancing and playing outside), and healthy nutrition (fruit, vegetables, tea without sugar and water). Even though fruit juice may be perceived as healthy [8], it is high in energy density and sugar content and could contribute to weight gain and overweight [8,38]. Because of this controversy, fruit juice was left out of analysis.

Children’s height and weight were measured by trained research assistants, using a measurement protocol. Children were measured during a physical education class, wearing light sports clothes and without shoes. Standing height was measured using the Seca 213 stadiometer (Seca, Hamburg, Germany), to the nearest decimal in centimetres. The Seca 803 digital weighing scale (Seca, Hamburg, Germany) was used to measure the child’s weight to the nearest decimal in kilogrammes. Children’s BMI z-score was calculated from their weight and height, while adjusting for their age and gender, using a Dutch reference population [39].

#### 2.4.3. Family Characteristics

Family characteristics were measured in the parent questionnaire, and consisted of postal code, level of education, family situation (partner/single) and Family Health Climate (FHC). The postal code was used to determine residential status score [35]. The residential status score ranged between −1.56 and 0.13 in the intervention region. The national mean residential status score is 0.2 [35]. Educational level of the parent was recoded into two categories [40]: (1) low (no education, primary school, secondary school, pre-vocational school or lower vocational education); and (2) high (higher vocational education or university). For two-parent families, a combined score was made, i.e., (1) low (both parents having a low level of education); (2) mixed (one parent low educated and one parent high educated); and (3) high (both parents having a high level of education).

In the parent questionnaire, parents filled in the validated Family Health Climate scale (FHC) [29], which was translated into Dutch [41]. The FHC is a 31-item questionnaire measuring shared family perceptions and cognitions concerning health behaviour, i.e., nutrition (FHC-NU) and physical activity (FHC-PA). The climate concerning PA and healthy nutrition is further divided into four and three concepts respectively, namely FHC-NU value, cohesion, communication and consensus, and FHC-PA value, cohesion and information. Items belonging to these concepts are statements, introduced by ‘In our family…’, and are answered on a four-point Likert-scale ranging from one (definitely false) to four (definitely true). FHC-NU value (four items) measures the importance of healthy nutrition within a family (‘… a healthy diet plays an important role in our lives’) (α = 0.62). FHC-NU cohesion (five items) includes the importance of eating together as a family (‘…everyone enjoys having meals together’) (α = 0.82). FHC-NU communication (five items) covers how normalised it is to talk about nutrition, and how family members support each other in eating healthy (‘…we remind each other to pay attention to a healthy diet’) (α = 0.82). FHC-NU consensus (three items) encompasses the level of agreement among family members concerning nutrition (‘…we usually agree on meals and food choices’) (α = 0.79). A high FHC-NU score implies a high value attached to healthy nutrition within a family. FHC-PA value (five items) comprises the value attached by all family members to being physically active (‘...it is normal to be physically active on a regular basis’) (α = 0.82). FHC-PA cohesion (five items) includes joint physical activities and experienced fun during such activities (‘…we enjoy exercising together’) (α = 0.88). Finally, FHC-PA information (four items) measures searching, sharing and using PA-related information as a family (‘…we read newspaper or magazine articles on fitness, physical activity and exercise’) (α = 0.80). The higher the FHC-PA score, the more integrated PA is within a family’s daily life [29].

### 2.5. Statistical Analyses

All results were analysed using SPSS 24.0 (IBM Corp., Armonk, NY, USA). Missing items of the FHC-PA and FHC nutrition scales (i.e., PA value, cohesion and information, and nutrition value, cohesion, communication and consensus) were imputed with the mean score of the other items of the same concept. Data were only imputed when a maximum of 10% of items per concept were missing. No other missing values were imputed.

Descriptive statistics were used to assess child and parent engagement and child and family characteristics of the intervention participants. Associations between percentage of challenges (per category) performed and child and family characteristics were assessed by conducting linear regression analysis.

First, the association between child and family characteristics as predictor variables and the percentage of challenges performed as outcome measures were analysed using bivariate linear regression analysis. The predictor variables used were the age, BMI z-score, gender, ethnicity (Western/non-Western), sports membership (member/non-member), preference PA and preference healthy nutrition of the child and the residential status score, the number of siblings, the combined educational level (low/mixed/high) and the FHC-PA and FHC-NU of the family.

Second, multivariate linear regression analyses were performed for the PA-related outcome measures (i.e., percentage PA child-only challenges, PA parental involvement challenges) by simultaneously using PA-related child characteristics as predictor variables in the model, i.e., age, gender, ethnicity, sports membership and preference PA. The same was done to assess the associations between the nutrition-related outcome measures (i.e., percentage nutrition child-only challenges, nutrition parental involvement challenges) and nutrition-related child characteristics, i.e., age, gender, ethnicity, sports membership and preference healthy nutrition. To assess the association between PA-related outcome measures and PA-related family characteristics, i.e., residential status score, number of siblings, parental educational level (low/mixed/high) and FHC-PA, predictor variables were entered simultaneously in this model while being corrected for all PA-related child characteristics. The same was done for the association between nutrition-related outcomes measures and nutrition-related family characteristics. A *p*-value of <0.05 was considered statistically significant.

## 3. Results

### 3.1. Child and Family Characteristics

In total, 406 children were eligible for inclusion in the study. Of these, 226 children (55.7%) participated in the study. Children had a mean age of 10.9 years (Table 2). Slightly more girls (55.8%) participated than boys, and more children were of Western ethnicity (62.7%). The majority of the children were members of a sports club (77.9%). Children preferred active activities slightly more (mean score: 15.7 out of 28). Regarding nutrition, there was a slight preference for unhealthy food and drink items (mean score: 7.5 out of 14). Of the parents, 186 (82.3% out of 226) returned their questionnaire. Mainly mothers filled in the questionnaire (82.9%), and most of them had a partner (84.4%). Almost half of the families consisted of two children (47.7%). In the majority of the families (50%), both parents had a low educational level and in 29.9% of the families, both parents had a high educational level. On average, families scored 2.8 on FHC PA, and 3.1 on FHC nutrition (range FHC score: 1 to 4).

### 3.2. Performance of Challenges

Of the 226 children participating in this study, all children (100%) performed at least one challenge and 211 children (93%) performed at least one parental involvement challenge (Table 3). Overall, PA challenges were performed more often than nutrition challenges. Only two children (0.9%) did not perform any of the PA-related challenges, whereas 85 children (37.6%) did not perform any of the nutrition-related challenges. Of the PA child-only challenges, the ‘Run as far as possible with your entire class, in 2 min’ challenge was completed by most children (*N* = 171, 75.7%) (Appendix B, Table A1). Of the PA parental involvement challenges, the ‘Come to school using a means of transport other than a car’ challenge was completed by most children (*N* = 176, 77.9%).

Regarding nutrition challenges, challenges involving parents were performed more often compared to child-only challenges, with 126 children (55.8%) performing at least one nutrition parental involvement challenge compared to 65 children, and 28.3%, performing at least one child-only nutrition challenge. Of the nutrition parental involvement challenges, ‘Make a work of art out of a piece of fruit’ was performed most often (34.5%), and of the nutrition child-only challenges ‘Find a healthy dinner recipe and bring this with you to school’ was performed most often (24.8%).

### 3.3. Associations between Child and Family Characteristics and Challenges Performed

Child characteristics significantly associated with the number of challenges performed were age, ethnicity and sports membership (Table 4 and Table 5). Younger children performed more challenges, specifically PA child-only challenges and nutrition challenges involving parents (Table 4). Children of Western ethnicity performed more PA parental involvement. Children who were a member of a sports club performed more PA child-only challenges.

Characteristics of the family significantly associated with the challenges performed by the child were residential status score, the number of siblings and a mixed parental level of education (Table 4 and Table 6). Children raised in high SES neighbourhoods (higher residential status score) were more likely to perform PA child-only challenges, and PA parental involvement challenges were performed more often by children of larger families. When compared to a low level of parental education, children of families with a mixed level of parental education completed more nutrition challenges, mainly child-only. This significant association was not found when comparing a mixed level of parental education to a high level of parental education.

## 4. Discussion

This study investigated the parent and child engagement in Challenge Me, an intervention targeting parents indirectly in a school-based intervention using PA- and nutrition-related challenges for children. Many children (*N* = 406) were exposed to the intervention. Of the 226 study participants, all (100%) participated in the intervention. Of these children, 93% performed at least one parental involvement challenge. This study showed that there was no difference in engagement in the intervention activities by children with a lower or higher BMI z-score, children with a higher or lower preference concerning active activities or healthy nutrition, or boys or girls. Also, children of families with an unhealthier or healthier climate concerning PA and nutrition did not differ in engagement in the intervention.

Nevertheless, other characteristics associated with the performance of certain types of challenges could be distinguished. For example, PA challenges were performed more often by children who were already a member of a sports club, and younger children. These active children might be more used to performing PA after school time and be more motivated to do so, and are thus more likely to perform the challenges compared to children who are not used to sports or PA activities that much. The fact that younger children performed more PA activities is not surprising, given the age-related decline in PA that is visible in children in high-income countries [42], which already starts at the age of seven [43].

Western children performed more PA parental involvement challenges compared to non-Western children. This is in line with previous research, showing that non-Western children perform significantly less PA (overall) than Western children [44,45,46]. However, additional analyses showed that this association between child ethnicity and the performance of PA parental involvement challenges was mediated by sports membership. The association between ethnicity and sports membership was significant (β = −0.23, 95% CI −0.31; −0.09, *p* < 0.01), with a higher percentage of children of Western ethnicity being a member of a sports club compared to children of non-Western ethnicity (83.0% versus 65.5%). Potentially, cultural differences between Western and non-Western families play a role in whether or not children participate in sports. It is unknown whether such cultural differences in PA and performance of PA and sports as a family exist in The Netherlands [43].

Our results also showed that children from larger families (i.e., having more siblings) performed more PA parental involvement challenges. Potentially, multiple siblings participated in the intervention, which meant a higher exposure to the challenges within the family. However, further research to support these findings needs to be done. Even though the FHC measures family cognitions on nutrition and PA behaviour specifically, FHC did not show any associations with parental involvement in nutrition or PA challenges performed. This could indicate that the challenges would be suited for all families, regardless of their current PA and nutrition family climate. However, both on FHC PA and FHC nutrition, families scored relatively high and variance was low, potentially limiting the probability of finding an association. Lastly, children living in higher SES neighbourhoods performed more PA child-only challenges compared to children living in lower SES neighbourhoods. This could be explained as lower SES neighbourhoods being perceived as less safe by parents [47], and could therefore be seen as less suited for children to perform PA on their own in their neighbourhood. Additionally, previous research has shown that high SES home environments are more supportive of PA in terms of availability of equipment [48]. We had expected that these factors would also influence the association between parental education level and the performance of PA challenges, with a higher performance of challenges in families of a high parental educational level, as educational level is an indicator of SES. However, we did not find children of high-educated parents performing more PA challenges (data not shown). We did see that children of families with a mixed educational level performed more nutrition child-only challenges compared to children of low-educated parents (but not compared to high-educated parents). However, we are unable to explain this association. 

Even though earlier studies often found a difference in performance of PA between boys and girls [45,49,50,51], our study did not show an association between gender and certain types of challenges performed. It can be that the gender-neutrality of the challenges played a role in this. While boys prefer activities like soccer and basketball, girls prefer activities like dancing, gymnastics and exercising to music [52]. However, none of the challenges had any of these elements incorporated in them. Gender-stratified analyses were performed for the association between child and family characteristics and challenges performed (per category), which showed a difference in key characteristics for boys and girls. For example, being a member of a sports club was significantly associated with the performance of PA child-only challenges for boys, yet this association was not significant for girls (data not shown). Interactions between variables and the performance of further sensitivity analysis were outside of the scope of this study, but should be studied further in the future to better understand who actively participated in the intervention and adjust the intervention accordingly.

Comparing the type of challenges performed, we found that in general, the children performed more PA challenges than nutrition challenges. Potentially, children regarded the PA challenges as more fun or as easier to perform. Children performed more or less the same percentage of PA child-only challenges and PA parental involvement challenges. By contrast, the nutrition challenges that were performed more often were the parental involvement challenges. Especially younger children performed more nutrition parental involvement challenges. To perform nutrition challenges (e.g., ‘Cook a healthy soup’), help from the parents was required: parents had to buy the ingredients or they needed to help with the preparation of the ingredients, e.g., the cutting of ingredients, which might be particularly true for the younger children in our study [16]. For the same reasons (e.g., parents buying groceries, and thus being in control of the food availability at home), nutrition behaviour of the child might be more dependent on and influenced by the parents than PA behaviour.

As a lower level of PA [10,53] and unhealthy nutrition [10] could lead to a negative energy balance, and consequently to overweight and/or obesity, an association between BMI z-score and performance of PA and healthy nutrition challenges was expected. However, BMI z-score did not affect the number of challenges performed, suggesting that Challenge Me would be suited for children of all weight statuses. 

Based on the results, it is recommended to take the above-mentioned child and family characteristics associated with certain types of challenges performed, i.e., child age, ethnicity, SES and family size, into account. For example, besides gender-neutral challenges, age-appropriate and culture-specific parent-child challenges might improve uptake of the intervention in the target group, including the vulnerable population. The intervention had potential to engage many children and parents, however, to achieve behaviour change, we hypothesize that a longer implementation period is needed. Intervention effects could be enhanced by promoting PA and healthy nutrition via various channels, and integrating the promotion of healthy behaviour into the standard school curriculum or daily routine [14]. Research is needed to study the effects of this type of indirect parental involvement interventions on children’s EBRBs, on children’s cognitive factors, like awareness and attitude, and on the home environment (e.g., parent-child play, healthier home climate).

### Strenghts and Limitations

To our knowledge, this is one of the few studies specifically aimed at investigating the engagement of children and parents in a school-based indirect parental involvement intervention. The reach of the intervention was extensive, and response rates among the participants high, with all children and over 80% of parents returning their questionnaire. Besides exploring the engagement in the intervention, we also aimed to specifically report on the characteristics of the children and parents participating in the challenges, as this information is currently underreported yet much needed to further understand and develop parental involvement interventions. Even though our study specifically reports on the characteristics of the study participants, we did not collect enough data to compare these characteristics to the characteristics of the school population and draw conclusions on the sample representativeness. To improve the match between the intervention and the characteristics, interests and needs of the children and their families, we suggest involving the target audience when developing the challenges [54]. This was not done when developing Challenge Me, resulting in some challenges being performed poorly. For example, ‘Make a vlog about healthy/unhealthy food in your surroundings’ was expected to be in line with children’s interest, but turned out to be an unpopular challenge.

Also, our study did not provide insights in the order in which the challenges were performed. All participating classes were free to determine the order in which they wanted to perform the challenges. However, the order in which challenges were introduced by the locally well-known person could have influenced children’s interest during the intervention, and subsequently their performance of the challenges. If children had already lost interest in the beginning, it is possible that later planned challenges could therefore have not been performed, regardless of their content and suitability to the target group. Challenge Me posters solely provided information on whether a challenge was performed or not, based on the evidence provided by a child to their teacher. However, it is unknown how strict teachers were in ticking off these challenges, and whether this varied between teachers. Also, it is hypothesised that teachers had a strong influence on the implementation of the intervention, and consequently on how enthusiastically children reacted to and engaged in the programme. For future research, it is recommended to gain more contextual insights in the implementation of the intervention, i.e., what was the influence of characteristics of teachers, whether children motivated each other to perform challenges, and what the uptake was of an indirect-parental involvement intervention in the home setting. It is particularly interesting to establish whether one parent, both parents or even the whole family (e.g., siblings and/or other caregivers) participated, and how they valued the intervention. Subsequently, future research should be done on the long-term engagement of children and parents in such interventions.

## 5. Conclusions

This study was an important first step towards gaining insight in a strategy to enhance parental involvement in energy balance-related interventions. The use of parent-child challenges has potential in increasing parental involvement in school-based interventions. All children participated in the Challenge Me intervention and 93% of the children had involved a parent, yet certain child and family characteristics should be taken into account when further developing parent-child challenges (such as the development of age-appropriate and culture-specific challenges) to ensure that all children and parents, including families of vulnerable populations, participate in these types of parental involvement interventions.

## Figures and Tables

**Table 1 ijerph-17-04838-t001:** Overview of ‘Çhallenge Me’ challenges.

	Child-Only	Parental Involvement
PA	Run as far as possible with your entire class, in 2 min. Try to perform as many bottle flips as possible in 1 min. Jump rope as long and often as you can, together with your PE teacher. Go to the town-/city centre and play a game of ‘the floor is lava’. Participate in an activity of your local youth organisation. Challenge a local youth worker for a battle. Challenge him/her in something you are good at. Watch instruction video: Free running.	Help your parent/guardian with the housekeeping. Ask your dad/mom/guardian/grandpa/grandma/neighbour what game they liked to play when they were young, and play that game. Go exercise for 30 min together with your parent(s)/guardian(s). Go walking, rollerblading, or stepping, together with your parent(s)/guardian(s). *Do a ‘bob for a job’ in your neighbourhood.* *Come to school using a means of transport other than a car.*
Nutrition	Find a healthy dinner recipe and bring this with you to school. Make a vlog about healthy/unhealthy food in your surroundings.	Cook a healthy soup. Make a healthy breakfast for someone in your household. *Make a work of art out of a piece of fruit.*

Note: PA = physical activity. Challenges are numbered at random, no order of completion was specified. Challenges in Italic require parental involvement dependent on the age of the child (older children are expected to be able to perform the challenge without help from the parent).

**Table 2 ijerph-17-04838-t002:** Child and family characteristics.

		*N*	%	Mean	SD
Child characteristics					
Age				10.9	1.0
BMI z				0.09	1.1
Gender	Boy	100	44.2		
	Girl	126	55.8		
Ethnicity	Western	141	62.7		
	non-Western	84	37.3		
Sports membership	Member	173	77.9		
	non-member	49	22.1		
Preference PA (0–28)	Active			15.7	3.3
Preference nutrition (0–14)	Healthy			6.2	2.8
Family characteristics					
Filled in by	Mother	141	82.9		
	Father	26	15.3		
	Other	3	1.8		
Family situation	Parent with partner	157	84.4		
	Single parent	29	15.6		
Residential status score				−0.9	0.8
Siblings	0	24	10.9		
	1	105	47.7		
	2	48	21.8		
	3	28	12.7		
	4–7	15	6.9		
Parental level of education	Low	92	50.0		
	Low/High mixed	37	20.1		
	High	55	29.9		
FHC-PA (1–4)				2.8	0.4
FHC nutrition (1–4)				3.1	0.4

Note: Numbers and percentages presented are based on complete cases. SD = standard deviation. PA = physical activity. FHC = family health climate.

**Table 3 ijerph-17-04838-t003:** Number of children who performed the challenges, per category.

	0 Challenges Performed (0%)		At Least One Challenge Performed (1–99%)		All Challenges Performed (100%)	
Category	*N*	%	*N*	%	*N*	%
PA	2	0.9	224	99.1	0	0
PA child-only (7 challenges)	17	7.5	205	90.7	4	1.8
PA parental involvement (6 challenges)	22	9.7	200	88.5	4	1.8
Nutrition	85	37.6	139	61.5	2	0.9
Nutrition child-only (2 challenges)	162	71.7	59	26.1	5	2.2
Nutrition parental involvement (3 challenges)	100	44.2	111	49.2	15	6.6

Note: *N* = number of children. PA = physical activity.

**Table 4 ijerph-17-04838-t004:** Bivariate associations between child and family characteristics and challenges performed.

		PA	PA Child-Only	PA Parental Involvement	Nutrition	Nutrition Child-Only	Nutrition Parental Involvement
Characteristic	Reference	β	β	β	β	Β	β
Age		**−0.153 ***	**−0.227 ***	0.016	**−0.154 ***	−0.063	**−0.165 ***
BMI z		−0.010	−0.064	0.065	0.048	0.036	0.042
Gender	Boys	0.047	0.052	0.018	0.098	0.009	0.122
Ethnicity	Western	**−0.151 ***	−0.102	**−0.146 ***	−0.078	−0.050	−0.074
Sports membership	Non-member	**0.226 ***	**0.221 ***	0.128	0.110	0.112	0.082
Preference active activities		0.091	0.073	0.073	N.A.	N.A.	N.A.
Preference healthy nutrition		N.A.	N.A.	N.A.	0.000	−0.055	0.030
Residential status score		0.046	0.079	−0.019	0.008	−0.041	0.032
Siblings		**0.142 ***	0.046	0.200 *	0.065	0.040	0.063
Parental educational level	(both) Low educated						
Mixed		0.095	0.091	0.054	0.167	0.147	0.139
(both) high educated		0.054	0.125	−0.065	0.125	0.130	0.094
FHC-PA		−0.021	0.038	−0.083	N.A.	N.A.	N.A.
FHC nutrition		N.A.	N.A.	N.A.	0.007	0.036	−0.010

Note: β = standardised beta coefficient. PA = physical activity. FHC = family health climate. N.A. = not applicable. * Significant result (*p* < 0.05) shown in bold print.

**Table 5 ijerph-17-04838-t005:** Multivariate associations between child characteristics and challenges performed.

		PA	PA Child-Only	PA Parental Involvement	Nutrition	Nutrition Child-Only	Nutrition Parental Involvement
Characteristic	Reference	β	β	β	β	β	β
Age		**−0.136 ***	**−0.208 ***	0.019	−0.131	−0.052	**−0.142 ***
BMI z		−0.008	−0.065	0.069	0.050	0.040	0.044
Gender	Boys	0.030	0.047	−0.004	0.115	0.040	0.127
Ethnicity	Western	−0.119	−0.056	**−0.148 ***	−0.052	−0.030	−0.052
Sports membership	Non-member	**0.182 ***	**0.180 ***	0.101	0.109	0.103	0.085
Preference active activities		0.095	0.070	0.084	N.A.	N.A.	N.A.
Preference healthy nutrition		N.A.	N.A.	N.A.	−0.032	−0.064	−0.007

Note: β = standardised beta coefficient. PA = physical activity. N.A. = not applicable. * Significant result (*p* < 0.05) shown in bold print.

**Table 6 ijerph-17-04838-t006:** Multivariate associations between family characteristics and challenges performed (adjusted for child characteristics).

		PA	PA Child-Only	PA Parental Involvement	Nutrition	Nutrition Child-Only	Nutrition Parental Involvement
Characteristic	Reference	β	β	β	β	β	β
Residential status score		0.082	**0.175 ***	−0.063	−0.063	−0.097	−0.029
Siblings		0.140	−0.002	**0.247 ***	0.091	−0.030	0.138
Parental educational level	(both) Low educated						
Mixed		0.075	0.036	0.090	**0.214 ***	**0.205 ***	0.169
(both) high educated		−0.067	−0.020	−0.094	0.133	0.142	0.097
FHC-PA		−0.002	0.047	−0.059	N.A.	N.A.	N.A.
FHC nutrition		N.A.	N.A.	N.A.	−0.038	−0.028	−0.035

Note: β = standardised beta coefficient. PA = physical activity. FHC = family health climate. N.A. = not applicable. * Significant result (*p* < 0.05) shown in bold print. Multivariate regression analysis adjusted for child characteristics: age, BMI z-score, gender (boy/girl), ethnicity (Wester/non-Western), sports membership (yes/no), preference active activities and preference healthy nutrition.

## Data Availability

The data that support the findings of this study are available upon reasonable request from the corresponding authors A.H.V. or D.H.H.V.K. The data are not publicly available because they contain information that could compromise research participant privacy/consent.

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
