# Peer review of "Involving Parents to Help Improve Children’s Energy Balance-Related Behaviours Through a School-Based Intervention"

_ijerph, 2020, doi:10.3390/ijerph17134838_

Round 1

Reviewer 1 Report

General comments:

While this paper is very well written, it lacks flow in the information provided. The purpose of the study was to examine 1) how many children and parents were reached by the intervention? and 2) which children and parents participated in the intervention according to child and family characteristics?

Based on the purpose of the study, I believe that the first question is very hard to answer because if all the schools invited accepted to take part in this study, the number of children will be different. Since the authors did not ask the children to participate in the first round, but asked the schools. Therefore, I don’t believe that the first question of this study is very well thought.

Regarding question 2: again, the children participated in the study was those who were invited to participate from 4 schools who accepted to take part in this study. So the answer to the question nr.2 is: the children who accepted to participate from those schools that was invited.

Furthermore, I don’t see clearly the purpose of the study.

Specific comments:

  1. Throughout the text, add coma (e.g.,) or (i.e.,).
  2. This sentence is very confusing: A sample of eight primary schools which participated in the larger intervention study, located in low SES neighbourhoods (based on scoring of the Netherlands Institute for Social Research [30]) in a city in the south of the Netherlands, were invited to participate in Challenge Me. Of these, four schools signed up for participation. Four schools declined to participate due to time constraints. Of the four participating schools, children from grades five to eight (aged 8 to 12 years) could participate in the intervention, and children from grades six to eight (aged 9 to 12) were eligible for inclusion in the study. Children were included when both parents had signed an informed consent form for participation in the larger study [29].???
  3. Line 116: together with or with???
  4. In the discussion part and for ethical purposes, I would strongly advise the authors to strictly hold to the notion of western and nonwestern.
  5. The author indicates that 8 schools from a major study was invited. Why from a major study? Or was the data from a major study?... unclear.

Unfortunately, while the study have valuable information and loads of data, it is confusing while reading. It has too much relevant information to the purpose.

Reviewer 2 Report

This is an exploratory study for a future school-based intervention. My comments follow:

  1. The authors talked about obesity epidemic in general in the introduction. It will be relevant to add information about obesity/childhood obesity in Netherlands where the study was conducted, and state the significance/meaning for such conducting studies in Netherlands.

  1. It is unclear how a sample of eight primary schools was selected. Convenience samples? Random sampling from a sample frame? Are these schools representative of the target population? This needs to be clarified.

  1. How were the eighteen PA and nutrition-related challenges developed? Are they evidence-based or from expert recommendations? Obesity intervention is a complex problem, and the evidence in terms of the effectiveness of diet and PA interventions remains mixed in the literature. The authors should justify valid potential benefits of these challenges.

  1. Analyses. It is unclear how missing data were handled. Were they just excluded from the analysis? Please clarify how much data were missing and justify/explain the influence of the missing data.

  1. Gender difference. It’s likely that different gender groups may have different patterns/behaviors. It would be informative to conduct stratified analysis by gender and see if there’s gender difference.

        6. In the discussion, how can future intervention be   improved/enhanced based on the results from this study should be better explored and some specific recommendations could be proposed.

Minor:

Abstract: It needs to be more specific and quantitative, for example, “Younger children, those who were a member of a sports club, those living in higher socioeconomic status (SES) neighbourhoods, children of Western ethnicity, and children from larger families performed more PA challenges. Younger children also performed more nutrition challenges involving parents.” Quantitative estimates should be added for “more”.

Reviewer 3 Report

It is good to see an intervention that provided a different way of introducing potential changes to diet and exercise in children. I only a few questions and several observations about the study.

P. 7 there is discrepancy between the total number of children and those that took part, but no explanation for it.

P. 3 Ethics - ethics was provided, however, there is no description of how parents and children gave informed consent.

 P. 3 Youth organisation helped to develop the challenges, however, this might bias the research to the interests of children attending the youth club. Inclusion of parents of children and children not involved might have provided different ideas. 

P. 8 It would be very interesting to follow up and evaluate the intervention to understand why some groups did not participate and why challenges were not completed, particularly the nutrition challenges. 

Round 2

Reviewer 1 Report

My comments were adressed by the Authors.

Reviewer 2 Report

Agreed to the revisions.